# Microscopic Analysis of the Wetting Morphology and Interfacial Bonding Mechanism of Preoxidised Kovar Alloys with Borosilicate Glass

**DOI:** 10.3390/ma16134628

**Published:** 2023-06-27

**Authors:** Jiajia Shen, Changjun Chen, Min Zhang

**Affiliations:** Laser Processing Research Center, School of Mechanical and Electric Engineering, Soochow University, Suzhou 215131, China; 20205229060@stu.suda.edu.cn (J.S.); zhangmin7822@suda.edu.cn (M.Z.)

**Keywords:** Kovar alloy, borosilicate glass, preoxidation treatment, wetting, microinterface analysis

## Abstract

This paper investigates the wettability of Kovar alloys with high-borosilicate glass and microscopically analyses the mechanism of wettability and diffusion between Kovar and borosilicate glass. First, Kovar was oxidised at 800 °C for 5, 15, 25, 35, and 60 min to observe the oxide morphology of the Kovar surface layer and to analyse the composition of the oxide layer. To investigate the wetting pattern formations of Kovar and high-borosilicate glass under different wetting temperatures, times, and preoxidation conditions, Kovar and high-borosilicate glass obtained from different oxidation treatments were held at 1060 °C for 20 min for wetting experiments, and the glass–metal wetting interface morphology and elemental distribution were observed using SEM and EDS. The elemental diffusion at the wetting interface between the borosilicate glass and the Kovar with different preoxidation and at the glass spreading boundary was investigated. The longitudinal diffusion of the liquid glass in the metal oxide layer formed a new tight chemical bond of Fe_2_SiO_4_, and the lateral diffusion of the liquid glass in the Kovar surface layer formed a black halo.

## 1. Introduction

With the continuous development of industrial production, the application of a single material in real-life industry does not apply, and two or more materials need to be used to meet specific structural and functional needs [1,2,3,4]. The sealing of glass and metal into a monolithic unit has the advantages of both materials in addition to the excellent properties of a single material, and the connection of glass and metal is needed in many fields, such as biomedicine, aerospace, vacuum-sealed devices, microelectronic packaging, semiconductors, and microelectromechanical systems [1,3,5,6,7,8].

Glass-to-metal sealing is a technique in which glass is melted and laminated to the metal being sealed. Due to the differences in the physicochemical properties of glass and metal and the fact that the glass-to-metal sealing process has been challenging [9], the most common method used today is to generate an oxide film on the surface of a Kovar alloy, which acts as a buffer layer to connect the glass to the metal [10,11,12,13,14]; however, the quality of the metal-to-glass sealing bond is influenced by several key parameters, such as the thickness, composition and microstructure of the oxide influences. A short preoxidation time to form a thin oxide layer can lead to weak chemical bonding; conversely, a long oxidation time to form a porous and sparse oxide layer can lead to seal failure. A thin oxide layer generated is approximately a few to a dozen microns, causing more difficultly for the study of the oxide layer on the glass-to-metal seal; therefore, precise control of the oxide thickness of the metal is essential for glass–metal sealing.

Although Kovar and glass sealing have been studied for decades, previous research has focused on borosilicate hard glass sealing with a low melting point and matching the coefficient of thermal expansion of Kovar [15,16,17,18,19], and less research has been performed on high-strength, high-melting-point, and high-borosilicate glasses with an even lower coefficient of thermal expansion of 3.3 × 10^−6^ K^−1^. To achieve a solid bond between high-borosilicate glass and metal, a chemical bond needs to be formed between the glass and the metal by mutual diffusion [20,21,22]. The current research on the spreading morphology of glass on the preoxidised metal surface during the wetting process and the diffusion bonding mechanism is not thorough enough, and this paper examines the role of oxide films for glass-to-metal sealing.

Therefore, in this study, the optimum oxidation temperature and holding time of Kovar alloys were investigated using a preoxidation method with a borosilicate glass matching sealer to examine the different formation patterns of oxide layers generated on the alloy surface. The bonding between different preoxidised Kovar alloys and high-borosilicate glass was then investigated, and further comprehensive microscopic analysis of the wetting and spreading morphology of Kovar and preoxidised high-borosilicate glass and the interfacial bonding mechanism was carried out based on the research by Chen [23].

## 2. Materials and Experimental Methods

### 2.1. Materials

Table 1 and Table 2 show the chemical composition of Kovar alloy (4j29) and borosilicate glass, respectively. The main elements of Kovar alloy are iron, nickel, cobalt and traces of manganese, silicon, copper, chromium and molybdenum, and the high-borosilicate glass mainly consists of SiO_2_ with small amounts of B_2_O_3_, (Na, K)_2_O and Al_2_O_3_.

### 2.2. Oxidation Treatment of the Kovar Alloys

The Kovar samples were thin sheets measuring 20 mm × 15 mm × 2 mm. Prior to the oxidation treatment, the metal surfaces were sandpapered and subsequently cleaned with an alcohol–acetone mixture. The samples were placed in an air-filled chamber furnace, heated from room temperature to 800 °C at a ramping rate of 10 °C min^−1^, held at five different isothermal holding temperatures for 5, 15, 25, 35 and 60 min, and then cooled to room temperature with the furnace at a ramping rate of 20 °C min^−1^.

### 2.3. Wetting Experiments

The high-borosilicate glass used in the wetting experiments was ground into glass powder in a ball mill, and the glass powder was compacted into a cylinder of 4 mm diameter and 4 mm height and horizontally placed in the centre of the upper surface of the preoxidised Kovar alloy, as shown in Figure 1. The sample was subsequently placed in a KF1400 chamber furnace. In the wetting experiments, the samples were heated from room temperature at a 10 °C min^−1^ ramping rate to different wetting temperatures and then cooled to room temperature with the furnace at a rate of 20 °C min^−1^.

### 2.4. Microscopic Observation and Elemental Analysis

The samples obtained from the above experiments were set with resin and subsequently ground and polished. The oxide film morphology and the glass–Kovar bond interface were then observed with an optical microscope (Motic BA310Met-T, Motic, Barcelona, Spain) and a scanning electron microscope (SEM, EVO-18, Carl Zeiss, Jena, Germany). The elemental distribution in the borosilicate glass–Kovar bond transition region was analysed using energy-dispersive spectroscopy (EDS). An X-ray diffractometer (XRD, Ultima IV, Rigaku Corp., Tokyo, Japan) was used to analyse the surface of the oxidised Kovar specimens and fractures at the glass–metal bond.

## 3. Experimental Results

### 3.1. Oxidation Experiment of the Kovar Alloy

Figure 2 shows the interfacial morphology of Kovar in the unoxidised and oxidised layers produced by holding at 800 °C for 5, 15, 25, 35 and 60 min; the resin is located in the black area on the left, the oxide produced is located in the grey area in the middle and the Kovar base material is located in the silver-grey area on the right. The unoxidised Kovar surface has no grey oxide layer and is closely adhered to the resin; at the same temperature, the oxide film on the Kovar surface gradually becomes thicker as the oxidation time increases. The thickness of the oxide film generated at 800 °C between 5 and 60 min of oxidation ranges from 4 to 16 μm, with a dense and uniform thickness occurring between 5 and 25 min; however, with the extension of time to 60 min, large pores form. This occurs because the oxide layer generated under this condition is fragile and not dense due to the long oxidation time; thus, falls off easily during the grinding and polishing process and eventually forms the hole shape seen in the figure.

Figure 3a shows the unoxidised Kovar alloy interface and elemental distribution curves. There was a significant difference in the elemental distribution at the interface due to the direct lamination of the resin to the metal. As shown in Figure 3b, the Kovar alloy oxidised at 800 °C for 25 min had two segmented areas in the middle of the resin and the Kovar alloy substrate according to the elemental curves: the main body of the oxide film and the Fe-depleted zone. The dense grey zone corresponded to the main part of the oxide film and was approximately 9 μm wide, with an enrichment of O and Fe and the presence of Co and traces of Ni. The area between the main part of the oxide film and the Kovar alloy base material was a loose intergranular oxidation zone of approximately 7 μm wide, with gradual decreases in Fe and O contents and an enrichment in Ni and Co contents. The area near the right side was the Kovar alloy matrix material, where the Fe content further increased. Based on the line scan spectra at the interface, the oxidised Kovar alloy surface had complex layers of oxides [24,25]; thus, the preoxidised Kovar alloy surface was divided into three regions from the outside to the inside (the Fe-rich oxidised layer body region, Fe-poor sparse region, and Kovar alloy matrix). Combined with Figure 4 and from the XRD physical analysis, after the oxidation treatment, the surface generated Fe_3_O_4_ and Fe_2_O_3_ with two main peaks. As the oxidation time was extended, heating or prolonged insulation led to the outwards diffusion of Fe in the base alloy due to the competition for Fe elements, resulting in the bottom of the alloy oxide layer appearing as a loose iron-deficiency band, and the final oxide layer embodied a multilayer structure.

### 3.2. Wetting Experiments with Borosilicate Glass and Kovar

#### 3.2.1. Analysis of Borosilicate Glass with Kovar Wetted Morphology

Figure 5a–h show the top view of the glass and unoxidised Kovar alloy wetted specimens held at 960 °C, 1020 °C, 1080 °C and 1140 °C for 10 min and 20 min, respectively, where the grey metal at the bottom is the wetted Kovar alloy substrate and the upper spherical crown is the high-borosilicate glass spread on the Kovar alloy surface after softening and solidification at high temperature. As observed from the photographs, the glass wetted specimen at 960 °C did not spread over the surface of the malleable alloy, indicating that the glass was not effectively wet with the malleable alloy since the high-borosilicate glass did not sufficiently soften at this temperature. When the temperature was 1020–1140 °C, as the temperature increased, the high-borosilicate glass gradually began to produce a good wetting effect with the Kovar alloy, a black halo formed around the glass spreading in this temperature range, and the range of the black halo further expanded as the temperature increased. Figure 5(a1–h1) show the corresponding side views, and the effect of temperature on the wetting shape and wetting angle can be observed more effectively by comparing the side views. The angle of wetting of the glass on the metal surface significantly decreased.

Figure 6a shows the trend of the spread diameter of high-borosilicate glass on the surface of the Kovar alloy with temperature. As shown in the figure, the glass powder uniformly compacted to a height of 4 mm and a diameter of 4 mm was placed in the centre of the metal surface. For the same holding time in the temperature range from 960 °C to 1140 °C with insulation for 20 min, the wetting spread diameter ranged between 4.1–8 mm, and at the same temperature range with insulation for 10 min, spread diameter was slightly larger. The 960 °C borosilicate glass just reached the softening temperature point. The glass in this state was unable to form an infiltration with the metal and therefore did not spread on the surface of the Kovar alloy. As the temperature increased, the diameter of the glass wetting and spreading on the metal surface became progressively larger in the range of 1020–1140 °C. Figure 6b shows the trend of the contact angle with temperature: the contact angle as a whole decreased with increasing wetting temperature, and the wetting angle decreased from 98.1° to 14.5° in the range of holding temperatures of 960–1140 °C for 20 min. At 960 °C, the wetting angles were all greater than 90°, and the glass was not wetted. As the temperature increased to 1080 °C during the process, the corresponding contact angle significantly decreased. Subsequently, the temperature continued to increase, and the decrease in wetting angle gradually flattened out when the wetting angle with holding for 20 min was smaller relative to the wetting angle with holding for 10 min at the same temperature.

To study the effect of holding time on the wetting behaviour of high-borosilicate glass and Kovar, Figure 7a–d shows the top view of high-borosilicate glass and unoxidised Kovar specimens wetted at 1040 °C for 5, 10, 15 and 20 min, and Figure 7(a1–d1) shows the corresponding side views. The spreading trend was not significant in the top or side views since a low wetting temperature and short holding time periods were selected, resulting in little variability in the spreading of the glass on the metal surface and the presence of black halos at different wetting times at this temperature. When the temperature was increased to 1080 °C with insulation for 5, 10, 15, and 20 min, the wetting angle change with respect to time was more evident; additionally, after increasing the temperature (Figure 7f,h), the Kovar alloy surface not covered by the glass wetting spread area of the oxide gradually began to fall off and exposed the deep black Kovar alloy base material. Based on the previous deductions on oxidation, clearly, the oxide film of the Kovar alloy was not dense and easily peeled off at higher temperatures; therefore, the area of the Kovar alloy surface not covered by borosilicate glass at 1080 °C was over-oxidised due to the high-temperature wetting process, resulting in excessive surface oxidation and falling off during the holding process. Although the control of temperature and time ensured that the high-borosilicate glass formed a good wetting effect with the Kovar alloy, the high-temperature conditions to achieve the softening and spreading of the glass could be very damaging to the surface of the metal.

Combined with the above experiments in the physical shape, Figure 8a,b show spreading diameter and wetting angle change curves for high-borosilicate glass at temperatures of 1040 °C and 1080 °C under different insulation times. In Figure 8a, the spreading diameter curve of the two temperatures from 0 to 20 min basically shows a linear upwards increase; however, the spreading diameter of wetting changes in the range of 1 mm, with a small time change difference, probably due to the small insulation time interval. The selection of a longer time leads to a more intense shedding phenomenon of the surface of the Kovar alloy; therefore, it was not suitable to expand the time for the wetting comparison experiments. Figure 8b shows the trend of the contact angle with time: the contact angle as a whole decreases with increasing wetting time. When the temperature is 1040 °C and the holding time is 5–20 min, the wetting angle decreases from 65° to 42°; however, when the temperature is 1080 °C and the holding time is 5–20 min, the wetting angle decreases from 41.8° to 30.5°. When the temperature is constant, as the wetting time increases, the wetting angle and spreading diameter change uniformly, indicating that the liquid glass flow is also more uniform in that time period and that the holding time is a key factor affecting the wetting angle.

To further investigate the differences in the wetting morphology of the different oxidation treatments of the fungible alloy and the high-borosilicate glass, combined with the discussion of the wetting temperature and time of the former, to achieve a fully wetted morphology, the wetting process was uniformly chosen for this experiment at a temperature of 1060 °C with a holding time of 20 min. Figure 9a–c1 shows the top- and side-view morphologies of the ferrous alloy and glass wetted specimens preoxidised at 800 °C for 5, 25 and 60 min, respectively. In the top view, the glass spread around the area with different degrees of elevated folds of oxide, and with the increase in preoxidation treatment time, the folds were more evident, mainly due to the different thicknesses of the oxide layer generated under the original different preoxidation conditions. After the wetting process, this was mainly because the areas not covered by glass during the wetting process were subjected to high temperatures again, resulting in further oxidation of the surface and the formation of raised folds. The wetting angles are very similar in size in the corresponding side views, indicating that the differences in the wetting spread of the oxide film on high-borosilicate glass at different thicknesses were not significant.

Combined with the physical morphology in the above experiments, the spreading diameters as well as the wetting angle curves of the high-borosilicate glass with different preoxidation treatments of the Kovar alloys are shown in Figure 10a,b, respectively. From Figure 10a, the spreading diameter of the Kovar surface of high-borosilicate glass preoxidised at 800 °C for different times ranges from 7.1–7.6 mm, indicating that the wetting spreading diameter of the Kovar surface gradually increases at the same temperature and time as the preoxidation time (i.e., the thicker the oxide layer generated) within the measurement error range; however, the increase is small, indicating that the thickness of the oxide film has an effect on the wetting spread diameter. Figure 10b shows the trend of the wetting angle of high-borosilicate glass with different preoxidation of the Kovar alloy surface. From the figure, high-borosilicate glass with 800 °C preoxidation at 5–60 min of the Kovar alloy surface forms wetting angles from 29.2° to 24.1°, indicating the wetting angle formed on the surface of the Kovar alloy at the same temperature and time (i.e., the thicker the oxide layer formed) decreases. Combined with the analysis of the preoxidation treatment of the previous section of the Kovar alloy, during Kovar alloy preoxidation at 800 °C for 0–60 min, the formation of the oxide layer thickness is 0–15 μm. In the wetting process of the glass in the oxide layer surface, the spreading diameter and wetting angle depend on the glass penetrating the depth of the oxide film; thus, a longer oxidation time correlates with a thicker oxide film and a smaller relative formation of the two wetting angles.

#### 3.2.2. Analysis of the Interface between Borosilicate Glass and Kovar Wetting

Glass and metal sealing quality also needs to be determined by good interfacial bonding, according to the previous research based on the wetting angle and spreading area under different oxidation times, temperatures and wetting macroscopic morphologies of different oxide layers. To further analyse the formation mechanism of microscopic bonding between high-borosilicate glass and different preoxidised Kovar alloys, the unoxidised and preoxidised Kovar alloys and high-borosilicate glass at 800 °C for 5 min, 25 min, and 60 min were held at 1060 °C for 20 min for wetting experiments, and the microscopic situation at the interface between the Kovar alloys and the high-borosilicate glass was observed after grinding and polishing using resin inlay.

The wetting and spreading of the high-borosilicate glass on the surface of the different preoxidised Kovar alloys is shown in Figure 11. In the figure, small and large holes inside the glass can be observed from the interface due to the interstices within the glass powder during the compaction process. The wetting experiments were carried out at the same temperature and for the same time, and the wetting angle of the spherical crowned high-borosilicate glass spread on the surface of the different oxidised frangible alloys slowly decreased with increasing thickness of the oxide layer, which verified the deduction of the previous investigation of the oxide layer on the wetting morphology. Figure 11b,c shows the bonding interface between the Kovar and the glass at 800 °C for 5 and 25 min, respectively. No gaps were observed in the bonding area.

To further investigate the interface between the glass and the different preoxidised Kovar alloys, local enlargements of the bonded parts at the above sections were made, as shown in Figure 12a–d, for the untreated and preoxidised at 800 °C for 5 min, 25 min and 60 min Kovar alloy–glass wetted interfaces, respectively. The dark-grey area at the top of the figure is the high-borosilicate glass, the light-grey part at the bottom is the Kovar alloy base material, and the middle area is the high-borosilicate glass wetted with the Kovar alloy to form different transition interfaces. The glass powder is compacted into a column, melted at high temperature, and then cooled and solidified to form a new glass body. The molten internal gas between the gaps in the glass powder cannot be released, resulting in a small number of pores forming on the glass side. In Figure 12a, due to the absence of the transition of the oxide layer, a small amount of reaction interface at the bonding during the wetting process is formed and is essentially a direct bond between the glass and the metal. For the Kovar alloy with a preoxidation of 5 min, the oxide layer is thinner; the transition zone formed in Figure 12b is narrower, but the thickness increases compared to the original oxide layer. For the Kovar alloy with a preoxidation of 25 min, the oxide layer is more suitable, and the transition zone formed is extremely dense, resulting in a very tight bond between the glass and metal, improving the bonding strength and ensuring a hermetic seal. The preoxidised layer at 800 °C for 60 min is thicker. Although the glass–metal transition zone is wider after wetting, the oxide layer formed under this condition is not dense, resulting in serious peeling of the glass–metal bonding area and causing the bonding to be vulnerable to failure. By comparing the different oxide layers with the wetted interface of the high-borosilicate glass, it is seen that the oxide layer plays a vital role in the glass–metal bond, and a small number of black spots are observed near the bottom of the transition area, with the black spots gradually increasing as the preoxidation time increases.

To further understand the microscopic characteristics and elemental changes at the interface in the intermediate transition region, the results of the analysis of the elemental distribution at the interface bond between the above four different oxidation thicknesses of the Kovar and high-borosilicate glass wetted states were combined with EDS surface scanning, as shown in Figure 13a–d.

From Figure 13a, the untreated fusible alloy is directly connected to the high-borosilicate glass wetting. According to the distribution of the main elements on both sides, no significant elemental fusion diffusion occurs at the junction between the two, only a very small amount of bonding is formed, and a small amount of Fe dissolves into the upper glass layer. Thus, the unoxidised fusible alloy and high-borosilicate glass wetting bonding process is only manifested as melted. The glass powder solidifies and adheres to the surface of the Kovar to form a mechanical bond.

Figure 13b,c show the surface scans of the wetted combination of the Kovar alloy and the high-borosilicate glass for 5 and 25 min, respectively. Additionally, there is also a small amount of Co in the transition region, which is likely to be present in the original oxide layer and not caused by the diffusion of Co from the lower fusible alloy during the wetting process. Ni is not involved in diffusion during the wetting process, and the distribution of Fe in the transition region is potentially due to the presence of Fe_2_O_3_ and Fe_3_O_4_ in the original oxide layer. At the bottom of the transition region, there is an iron-poor zone similar to the bottom of the oxide layer. The distribution of Fe in the wetting and in the oxide layer is consistent, indicating that the existing transition region is the original oxide region, but it is clearly thicker than the thickness of the original oxide layer. At the bottom of the original oxide layer is an Fe-poor region, and there is a small amount of black-spot material in the bottom layer. The formation of black spots is attributed to the penetration of the glass into the grain boundaries of the oxide layer within the Kovar alloy forming a new material Fe_2_SiO_4_. This material combined with the diffusion of Si indicates that the glass on the upper side penetrates through the main oxide layer during the wetting process and then diffuses further along the original intergranular oxide into the underlying Fe-poor region. In summary, from the microscopic diffusion point of view, the entire wetting process is predominantly caused by the diffusion of Si in the molten glass towards the interior of the Kovar alloy by means of the oxide layer under the action of high temperature. Other elements are not involved in the wetting reaction, and the oxidised Kovar alloy and the unoxidised Kovar alloy wetting conditions are very different. The glass and oxidised Kovar alloy form a chemical bond, which further confirms that the oxide layer can promote the diffusion and fusion of glass and metal. This further demonstrates the ability of the oxide layer to promote diffusion and fusion of the glass to the metal, playing a vital role in the transition to the glass-metal seal.

Figure 13d shows the EDS surface scans at the wetted interface between the preoxidised 60 min Kovar alloy and the high-borosilicate glass. Due to the thick and porous oxide film oxidised at 800 °C for 60 min, the bonding area is extremely fragile after wetting with the glass, and a large amount of peeling occurs during the grinding process. The elemental diffusion mechanism does not differ from the 5 and 25 min oxide film wetting, mainly because Si in the molten glass diffuses in the fragile oxide layer and the dense distribution of intergranular oxides under this long oxidation causes more diffusion in the bottom glass layer of the transition region.

To clarify the extent of mutual wetting diffusion, the above wetting interface was used to perform EDS line scan analysis, and the results are shown in Figure 14a,b. In Figure 14a, the main element distribution curve shows that there was no oxide layer between the two layers for bond formation, resulting in a large difference in the distribution trend of the main elements, such as Fe, Co, Ni and Si, between the two bonding regions. This result further indicated that the bonding between the two was caused only by the glass melting and spreading on the metal surface to form a physical bond. Figure 14b shows the morphology and line scan trend of the borosilicate glass wetting of the fusible alloy after 25 min of preoxidation at 800 °C. Combined with the deductions of the previous section on oxidation, the fusible alloy formed after 25 min of oxidation at 800 °C generated an oxide film of approximately 9 μm, but after wetting, a new chemical reaction occurred during the diffusion process to produce a new transition with a thickness of approximately 20 μm in the middle. The thickness of the new transitions is significantly thickened by approximately 20 μm. Clearly, Si is gently and uniformly distributed in the intermediate transitions.

#### 3.2.3. Study of the Wetting Boundary and Black Halo Formation between High-Borosilicate Glass and Kovar Alloys

Based on the analysis of the outer seal edge interface at the glass–metal interface, the wetting response at the ends of the glass spreading was more intense than at the inner glass–metal bond away from the edge, and this region was considered to be the key location for seal failure [6]. During the wetting process and in conjunction with the effect of previous process parameters on the morphology, the high-borosilicate glass formed a black halo at the spreading edge of the Kovar surface, which is closely related to the outer sealing edge of the glass–metal interface; therefore, it was necessary to investigate the morphology and diffusion at the wetted edge to understand the mechanism of black halo formation.

Figure 15 shows the interfacial morphology and EDS surface scans at the wetted and spread edges of the Kovar alloy and high-borosilicate glass after oxidation treatment at 800 °C for 25 min. The Kovar alloy substrate is located at the bottom and the spread edges of the high-borosilicate glass after wetting and solidification are located at the top right. The transition zone formed at both ends of the edge is more reactive than the transition zone at the glass–metal combination in Figure 13c, and the thickness is wider than 50 μm. Thus, during the wetting process, the two ends of the borosilicate glass that are not covered by the borosilicate glass come into contact with the furnace atmosphere under the action of high temperature and gradually grow upwards. The oxidation growth, accompanied by the process of glass softening, leads to the spreading of the liquid glass along the growing oxide film. The distribution of Si at the transition zone at the wetted edge shows that there is diffusion in the horizontal direction in the transition zone. From the morphology diagram, cracks form on the glass side at the wetting boundary between the Kovar and the high-borosilicate glass due to the difference in the coefficient of thermal expansion between the physical properties of the glass and the metal. The coefficient of thermal expansion of the high-borosilicate glass is 3.3 × 10^−6^ K^−1^, which differs from the coefficient of thermal expansion of the metal oxide layer. While cooling with the furnace, the liquid high-borosilicate glass and the surface of the Kovar cannot simultaneously shrink. This results in residual stress after the glass is cooled, solidified and spread on the metal surface. When the stress exceeds the bearing range of the glass, cracks are formed.

#### 3.2.4. Study of the Fracture and Composition of High-Borosilicate Glass with Frangible Alloys

To derive the physical phase composition of the wetted interface between the high-borosilicate glass and the different surface-treated Kovar alloys as well as the bonding mechanism, the wetted bond was peeled apart using shear force, and the macroscopic morphology of the fracture is shown in Figure 16. Figure 16a shows an SEM image of the top view of the wetted bond interunit. The bulbous crown in the middle is high-borosilicate glass, and the black halo is clearly observed on the outside of the glass [16,17]. After shear separation, Figure 16b,c show the fractures on the metal side and the glass side, respectively. Both the glass and metal fractures show a honeycomb fracture surface, with a small amount of residual material on the edge of the fracture surface of the Kovar alloy.

The fracture at the lower left edge of Figure 16b was selected for EDS analysis, as shown in Figure 17. In the figure, Si and Fe are concentrated in the internal part of the fracture. According to the element distribution at the previous interface, Si and Fe are enriched together in the transition area where the glass is combined with the metal, indicating that internal fracture occurs in the transition area. In the black halo and fracture junction position, Si is enriched with no Fe. Combined with Figure 14 showing the glass spreading in the metal at both ends of the formation of fractures, the section obtained after shear stress shows a brittle fracture along the edge stress generated at the fracture; thus, this region is the glass residue. The dark area in the middle is the black halo at the edge of the glass spreading. The distribution of the corresponding elements shows that there is a small amount of Fe and Co in this area, accompanied by a large amount of Si. This result confirms the inference of the cause of the black halo at the wetting edge, indicates that the spreading of the high-borosilicate glass towards the ends during the wetting process leads to the diffusion of Si in the horizontal direction on the surface of the oxide layer generated later, and further shows the nature of the black halo.

In order to find out whether new phases were formed at the interface between the borosilicate glass and the kovar alloy during the wetting process, XRD phase analysis was carried out on the pre-oxidised kovar alloy and the fracture on the borosilicate glass side (Figure 16b,c above), respectively, and the XRD diffraction pattern is shown in Figure 18. According to the XRD physical phase analysis results, it was found that the high borosilicate glass side fracture phase mainly contains SiO_2_ and a small amount of Fe_2_SiO_4_, however the kovar alloy side fracture composition also mainly contains SiO_2_ and Fe_2_SiO_4_, this is because the wetting bond separates a small amount of high borosilicate glass residue in the fracture side of the kovar alloy after shearing, so there will be SiO_2_ present, during the wetting process During the wetting process new chemical bonding reactions are formed in the transition region between the borosilicate glass and the kovar alloy after elemental diffusion, so that a new phase Fe_2_SiO_4_ is formed on the glass and metal side of the bond. small amounts of Fe_3_O_4_ and Fe_2_O_3_ are also found on the fracture side of the kovar alloy, which are components of the original oxide film.

### 3.3. Discussion

#### 3.3.1. Mechanism for the Formation of High-Borosilicate Glass in the Wetted Form of Kovar Alloys

Surface tension γ is defined as the energy required to increase the surface area of a liquid by a specific amount. It is one of the very important thermophysical properties of liquids and determines pertinent information about the impact of the strength of the interaction between the liquid and the solid; the surface tension is the most important influencing factor reflecting the wetting and spreading of glass on metal surfaces. During the wetting process, as the temperature increases, the softening temperature of the glass is gradually reached, resulting in a reduction in the viscosity of the liquid glass. This process affects the surface tension of the liquid phase of the glass. The empirical equations of Eötvos [26] were used to analyse the temperature dependence of the surface tension and molar volume of liquid glasses during the wetting of borosilicate glasses with Kovar.
(1)$$\gamma {\left(V\right)}^{2/3}=K(T_{c}-T)$$

In Equation (1), γ is the surface tension, V is the molar volume of the liquid g.cm^−3^, K is Planck’s constant, and T and T_c_ are the Kelvin temperature and critical temperature, respectively. The equation shows that surface tension is a linear function of temperature and decreases as the temperature increases. Similarly, in the wetting process, as the temperature increases, the liquid–gas interfacial tension of the liquid glass decreases for the wetting and spreading process of the liquid glass on the metal surface.

Solid–liquid droplet experiments have been widely used to study solid–liquid interfaces. The interfacial morphology by wetting and flowing of glass on a metal surface forms from the interaction between the interfacial energies of the gas, liquid and solid phases. According to the interfacial energy (γ) relationship, the wetting phenomenon can be expressed by the Young–Dupré equation [27], as follows:$$\gamma_{S/V}-\gamma_{S/L}=\gamma_{L/V}\mathrm{Cos}\theta$$
where S/V denotes the solid–gas interface, S/L denotes the solid–liquid interface and L/V denotes the liquid–gas interface. The angle at the three-phase boundary passing inside the liquid to the gas–liquid interface is called the contact angle, expressed as θ. Generally, the magnitude of the contact angle θ is a macroscopic determination of good or bad wettability. The whole wetting process is shown in Figure 19. As the temperature increases, the molten glass mobility increases, resulting in a reduction in tension (interfacial energy) between the gas and liquid phases, and the change in interfacial energy between the glass and the Kovar alloy becomes the driving force for wetting (γ_s/L_ − γ_s/v_). When the temperature reaches the softening temperature of the glass in the beginning (90°< θ < 180°), the liquid glass is not wetted with the metal and not able to effectively spread on the metal surface, there is a small spreading and fitting area between the two, and no wetting occurs. As the temperature and holding time increase, the surface tension of the liquid glass continues to decrease, enhancing the liquidity of the molten glass. This driving force exceeds the energy of the liquid glass surface; specifically, γ_S/L_ < γ_S/V_ when the wetting angle range gradually narrows in the range of 0 < θ < 90 degrees, thus reflecting that the liquid can wet the solid. Thus, the molten glass spreads on the metal surface. These processes help in sealing and joining.

In the borosilicate glass and Kovar wetting system, the change in interfacial energy [26] due to the temperature and time of holding causes the wetted liquid glass to gradually become a flattened spherical crown shape and eventually forms a spreading shape when the glass form reaches equilibrium, cools, and solidifies. The spreading of the ends of the liquid glass during the atmospheric wetting process also results in the formation of special structural bodies. Based on the analysis of the cross-sectional morphology of the wetted interior and the edges of the ends and in conjunction with the theory of reoxidation growth of the Kovar [14], the entire wetting process of the high-borosilicate glass and the Kovar can be explained in three stages, as shown in Figure 20.

In the first stage (I–II), the heating process begins when the glass powder is compacted into a column and placed on the surface of the preoxidised Kovar alloy. When close to the glass softening temperature (≈850 °C), the compacted glass powder covers the surface of the Kovar alloy, resulting in the covered area not being able to come into contact with the atmosphere inside the furnace; the area not covered by the glass powder is exposed to the atmosphere. Thus, the initial process of oxidation on the surface of the Kovar alloy continues as the temperature increases to the glass softening temperature, resulting in the growth of the oxide layer on the metal surface in the uncovered area, and no black halo has formed at this stage.

In the second stage (II–III), after the softening temperature of the glass is reached, the liquid glass begins to soften and flow, driven by thermodynamics as well as gravity; the oxidation continues to grow on the uncovered Kovar alloy surface at both ends. Thus, the liquid glass begins to flow along the growing oxide layer and moves towards the Kovar alloy. At the bonding site, the liquid glass adheres to the preoxidised Kovar surface and simultaneously begins to diffuse into the oxide film. At this stage, the softened glass begins to fill jagged microstructure pits, and a black halo gradually forms at the edge of the glass spread; however, the wetting angle between the glass and metal in this state is greater than 90°, and no wetting of the two occurs.

The third stage of the wetting process steadily approaches the temperature and time needed for wetting and gradually reaches an equilibrium state with the glass spreading over the metal surface. At the ends of the metal not covered by glass, the surface morphology of the previously prepared laser microstructure is completely covered by the oxide film generated later. Finally, as the temperature decreases, the liquid glass takes on a spherical crown shape and solidifies firmly on the surface of the Kovar alloy, eventually forming a characteristic morphology at both ends of the interface, as shown in Figure 20IV.

In summary, the formation of the wetting profile of the high-borosilicate glass on the surface of the Kovar alloy is mainly attributed to the thermodynamically driven effect of changing the interface energy by adjusting the wetting temperature and holding time to enable the liquid glass to flow, and the entire wetting process is accompanied by oxidation of the metal surface, eventually showing a unique macroscopic wetting profile.

#### 3.3.2. Wetting Microdiffusion of High-Borosilicate Glass with Kovar and Composition Formation Mechanisms

In the macroscopic morphology, the borosilicate glass gradually adheres to the surface of the Kovar in a flattened spherical crown shape in the wetting mechanism. The presence of a black halo can be clearly observed in the pictures at various temperatures and times, and in the wetting interface diagrams of the Kovar alloy and the high-borosilicate glass with different surface treatments only when the borosilicate glass forms a wetting angle of less than 90° on the surface of the Kovar alloy. This result indicates that the presence of a black halo is to some extent explained by the reaction wetting mechanism but not by the macroscopic morphological wetting mechanism.

In our study, the black halo formation is relatively small. In the study of copper and silicate glass wetting, Zhang Min et al. [17] found that under different temperatures and times, the edge of the spherical crown-shaped glass showed an evident halo phenomenon. It was believed that the rapid adsorption and thin layer flow mechanism could explain the formation of black halo. In the study of borosilicate glass and borosilicate alloy wetting, Luo et al. [16] also found the black halo phenomenon. This black halo found by Luo when studying the wetting of borosilicate glass with Kovar was thought to be a result of elemental diffusion and the formation of different viscosities in different states of liquid glass. The existing mechanisms for the formation of black halos do not adequately explain the actual cause of the halo formation. Based on the analysis of the interfacial morphology and the distribution of EDS elements at the wetting ends in conjunction with the capillary driving force theory to discuss the mechanism of diffusion at the glass–metal interface and the formation of the wetting black halo at the microscopic level, Loudj [28] concluded that during the grain growth as a transport mechanism, solutes diffuse and migrate along the grain boundary dragging process. Based this concept, Si is able to diffuse and migrate in the oxide layer during the wetting process. In combination with the previous microscopic interfaces and the distribution of elements in the fracture, the glass diffusion mechanism during the wetting process can be divided into two aspects: longitudinal and transverse diffusion. The combination is shown in Figure 21.

(1) Longitudinal diffusion: During the softening process, the borosilicate glass diffuses by osmosis at the interface between the two, and the area of diffusion includes the oxide layer generated by preoxidation as well as the loose porous area at the base of the oxide layer. In combination with the previous wetting interface analysis, the liquid glass initially dissolves and penetrates the oxide layer such that the evident Si diffusion in the middle forms a transition region; in combination with the corresponding line scan, the penetration depth of the glass depends on the thickness of the oxide layer at a sufficient wetting temperature and time. When the glass is dissolved and saturated in the oxide layer to form the transition region and combined with the preoxidation, the bottom layer of the Kovar oxide layer is a loose and porous Fe-depleted region, further creating conditions for the glass to diffuse into the interior of the Kovar; therefore, the glass diffuses into the original intergranular oxide at the bottom of the oxide film. For the unoxidised Kovar and high-borosilicate glass wetting, there is no mutual integration of the two diffusion processes, and the bonding area is simply due to the solidification of the glass adhering to the surface of the Kovar. Combined with the previous deductions on the wetting morphology, the original jagged grooves of the high-borosilicate glass wetted with the microstructured Kovar were filled with liquid high-borosilicate glass, and elemental Fe was found within the filled pits, indicating that the metallic elements within the bonding site also diffuse into each other within the grooves.

(2) Lateral diffusion: The lateral spreading of the Kovar alloy wetting in the high-borosilicate glass is similar for all different surface treatments. The glass diffuses laterally in the slope of the oxide layer growing along the ends such that a uniform amount of Si is present in the transition region of the wetting interface at the corresponding ends. The end point of the lateral diffusion is not the boundary of the spreading morphology of the liquid borosilicate glass, but the glass diffuses further to the sides in small amounts along the post-growth oxide film under capillary driving forces. This is clearly demonstrated with the previous interface morphology at the wetted edges and the EDS data, culminating in the formation of the black halo at the boundary by the lateral diffusion.

Thus, the liquid borosilicate glass is continuously dissolved in the sparse Kovar oxide layer, leading to lateral and longitudinal diffusion of the glass and ultimately to the formation of a firm bond. The formation of a special black halo in the wetting transverse diffusion further indicates that the preoxidation of the Kovar alloy plays a crucial transitional role in the bonding of the glass to the Kovar alloy.

## 4. Conclusions

The oxidation of the Kovar alloy was carried out at 800 °C for different times to obtain different thicknesses of the oxide film. The wetting experiments were performed using the Kovar alloy with high-borosilicate glass with different oxide layers obtained to summarise the borosilicate glass–Kovar alloy wetting mechanism, and the following conclusions were drawn.

(1)The oxide film generation thickness was 0–15 μm at 800 °C oxidation in the range of 0 to 60 min, and the thickness due to oxidation was denser in the range of 0 to 25 min; however, with the extension of time to 60 min, the oxide layer exhibited a loose structure and large pores. The oxide layer was divided into three regions according to the Fe element distribution (the Fe-rich matrix of Kovar alloy, Fe-poor porous region and Fe-rich oxidation region), and the main components of the oxide layer were Fe_3_O_4_ and Fe_2_O_3_.(2)The wettability of the Kovar alloy and high-borosilicate glass depended on temperature and time. As the temperature and holding time increased, the area of borosilicate glass spread on the surface of the Kovar alloy increased, and the wettability angle between the two decreased; however, a high temperature and time increase tended to damage the surface layer of the Kovar alloy.(3)The wetting mechanism of high-borosilicate glass on the surface of the Kovar alloy was mainly attributed to mechanical bonding. The entire wetting process of liquid glass filling the gap between the glass and the metal was accompanied by the reoxidation process on the surface of the Kovar alloy and the flow of liquid glass driven by thermodynamics to form the final macroscopic wetting profile.(4)The glass-to-metal wetting mechanism was divided into two aspects: longitudinal and transverse diffusion. The transverse diffusion of glass in the oxide layer of the Kovar alloy caused the formation of the black halo. During the wetting process, the high-borosilicate glass diffused longitudinally at the Kovar oxide interface under the action of capillary driving forces to form new chemical bonds, resulting in a new substance, Fe_2_SiO_4_, and the glass diffused further along the intergranular oxide towards the interior of the Kovar alloy.

## Figures and Tables

**Figure 1 materials-16-04628-f001:**
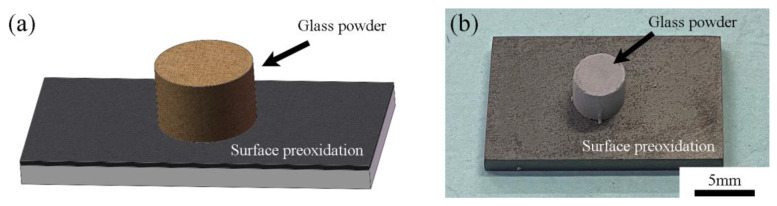
(**a**) Schematic diagram of oxidative wetting and (**b**) physical picture.

**Figure 2 materials-16-04628-f002:**
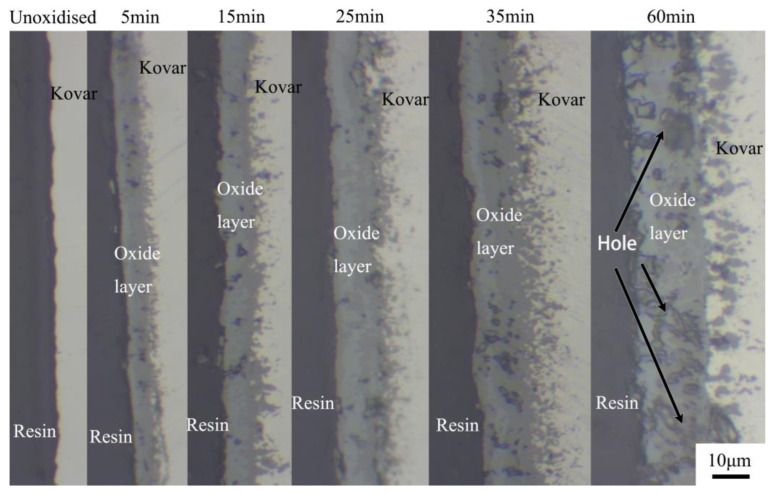
Images of the sectional morphology of the unoxidised and preoxidised samples at 800 °C for 5, 15, 25, 35, and 60 min.

**Figure 3 materials-16-04628-f003:**
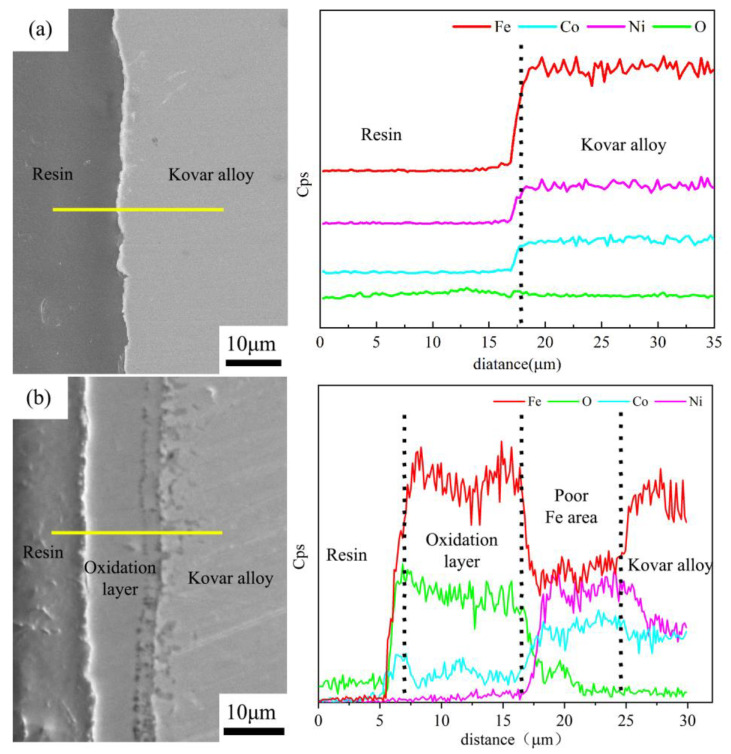
EDS line diagram of Kovar: (**a**) unoxidised and (**b**) oxidised.

**Figure 4 materials-16-04628-f004:**
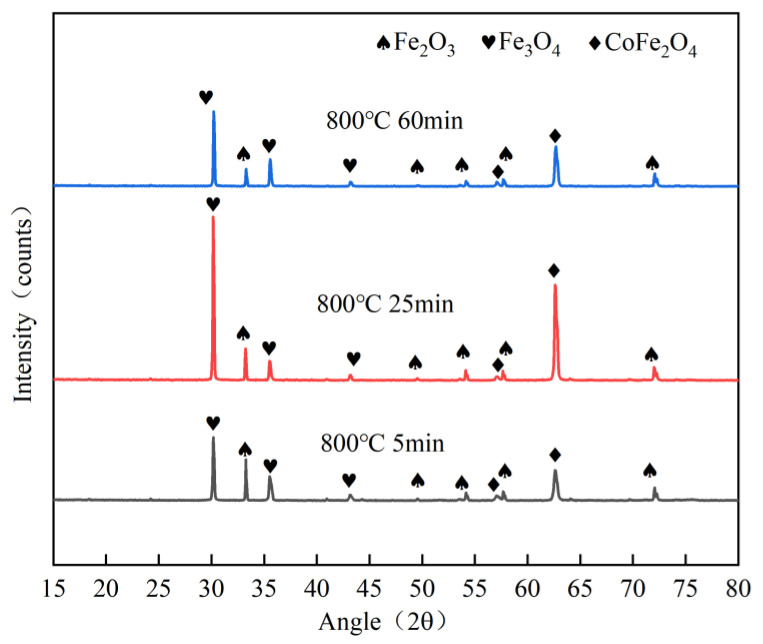
XRD patterns of preoxidised samples at 800 °C for 5, 25, and 60 min.

**Figure 5 materials-16-04628-f005:**
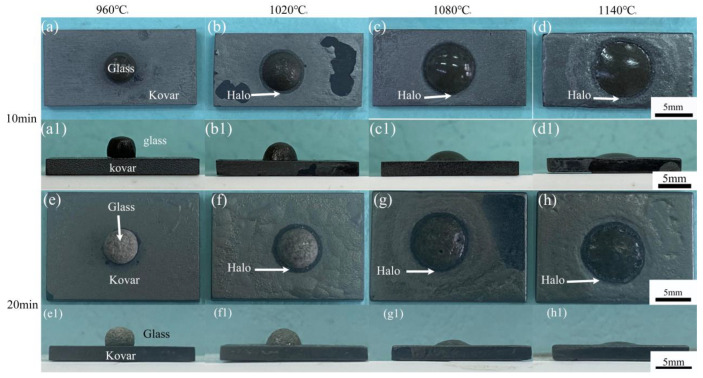
Top and side views of the high-borosilicate glass wetted with Kovar alloy at different temperatures: (**a**–**d**) insulated 960–1140 °C for 10 min, top view; (**a1**–**d1**) insulated 960–1140 °C for 10 min, side view; (**e**–**h**) holding temperature 960–1140 °C for 20 min, top view; (**e1**–**h1**) holding temperature 960–1140 °C for 20 min, side view.

**Figure 6 materials-16-04628-f006:**
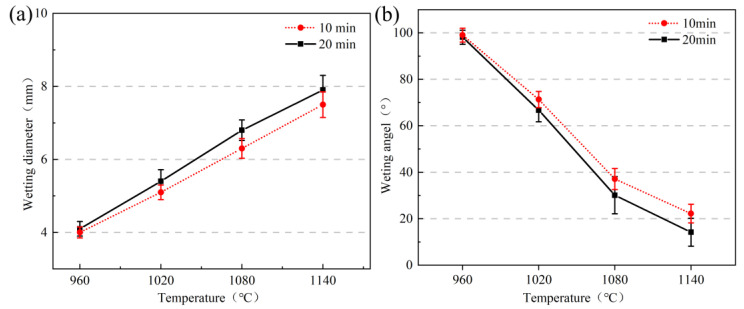
Spread diameter and wetting angle of the high-borosilicate glass on the surface of Kovar alloys at different temperatures: (**a**) spreading diameter; (**b**) wetting angle.

**Figure 7 materials-16-04628-f007:**
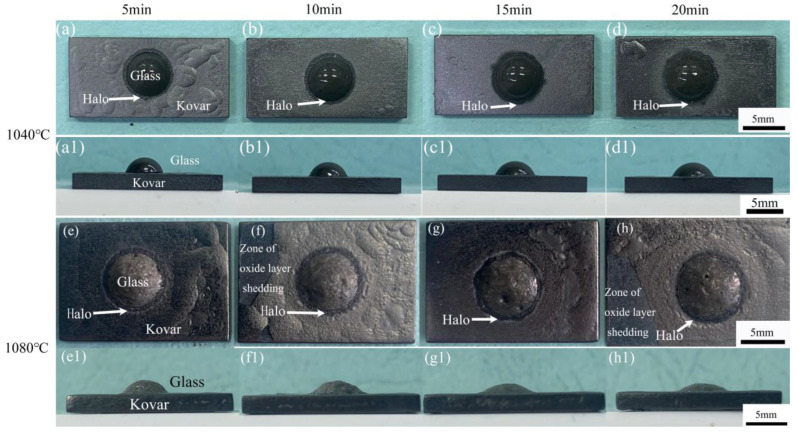
Top and side views of high-borosilicate glass wetted with the Kovar alloy at different times: (**a**–**d**) insulation at 1040 °C for 5–20 min, top view; (**a1**–**d1**) insulation at 1040 °C for 5–20 min, side view; (**e**–**h**) holding temperature of 1080 °C for 5–20 min, top view; and (**e1**–**h1**) holding temperature of 1080 °C for 5–20 min, side view.

**Figure 8 materials-16-04628-f008:**
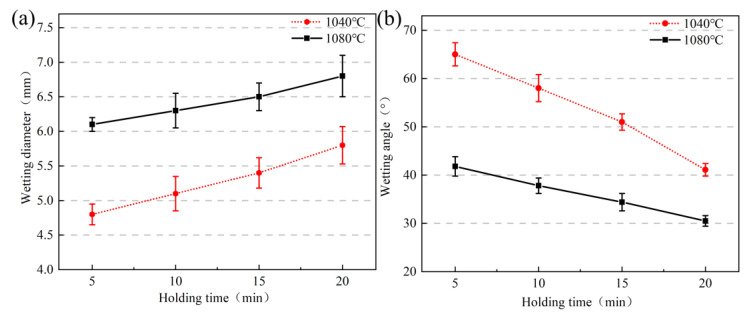
Spread diameter and wetting angle of high-borosilicate glass on the surface of Kovar alloys at different times: (**a**) spreading diameter and (**b**) wetting angle.

**Figure 9 materials-16-04628-f009:**
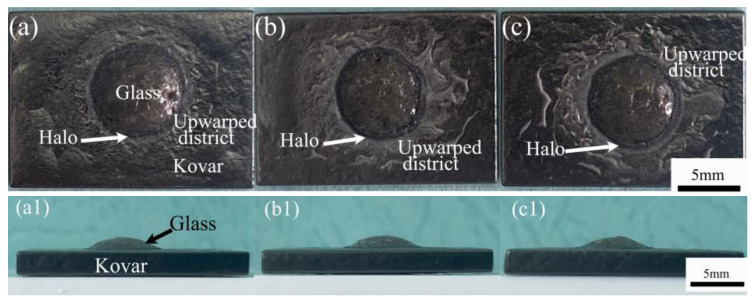
Top and side views of different oxidation treatments of the Kovar alloy with high-borosilicate glass wetting: (**a**–**c**) top view of holding at 800 °C for 5, 25, and 60 min and (**a1**–**c1**) side view of holding at 800 °C for 5, 25, and 60 min.

**Figure 10 materials-16-04628-f010:**
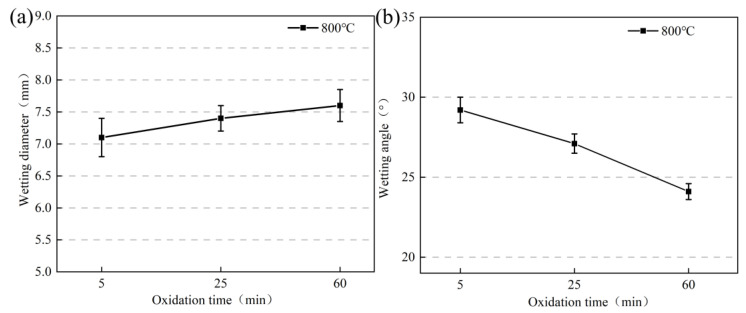
Spread diameter and wetting angle of high-borosilicate glass on the surface of Kovar alloys under different oxidation times: (**a**) spreading diameters and (**b**) wetting angles.

**Figure 11 materials-16-04628-f011:**
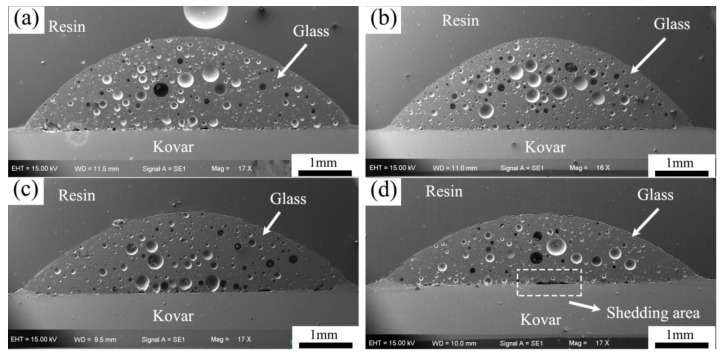
Wetting interface between borosilicate glass under different Kovar oxidation times. (**a**) no oxidation; (**b**) 800 °C for 5 min; (**c**) 800 °C for 25 min; (**d**) 800 °C for 60 min.

**Figure 12 materials-16-04628-f012:**
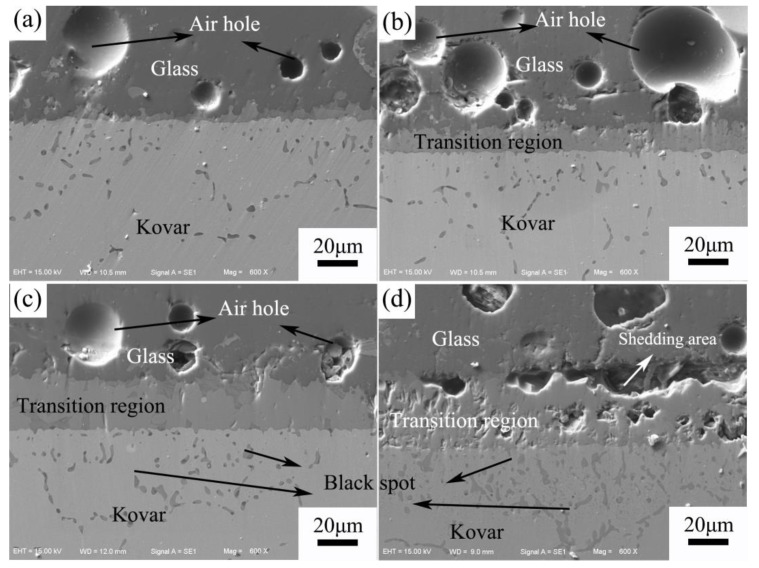
Wetting interface between borosilicate glass under different Kovar oxidation times. (**a**) No oxidation; (**b**) 800 °C for 5 min; (**c**) 800 °C for 25 min; (**d**) 800 °C for 60 min.

**Figure 13 materials-16-04628-f013:**
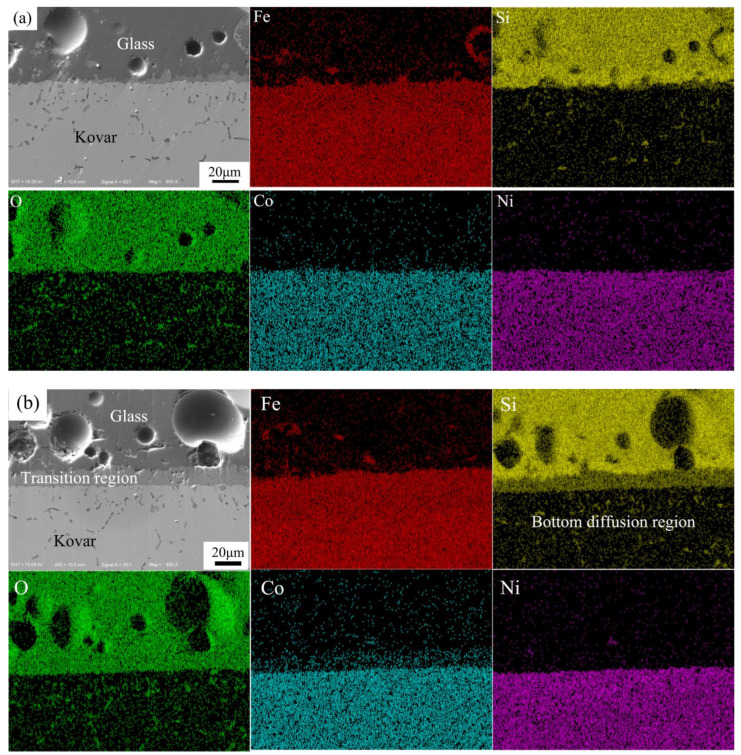
EDS spectrum of borosilicate glass wetting with different oxidation reactions of Kovar. (**a**) No oxidation; (**b**) 800 °C for 5 min; (**c**) 800 °C for 25 min; (**d**) 800 °C for 60 min.

**Figure 14 materials-16-04628-f014:**
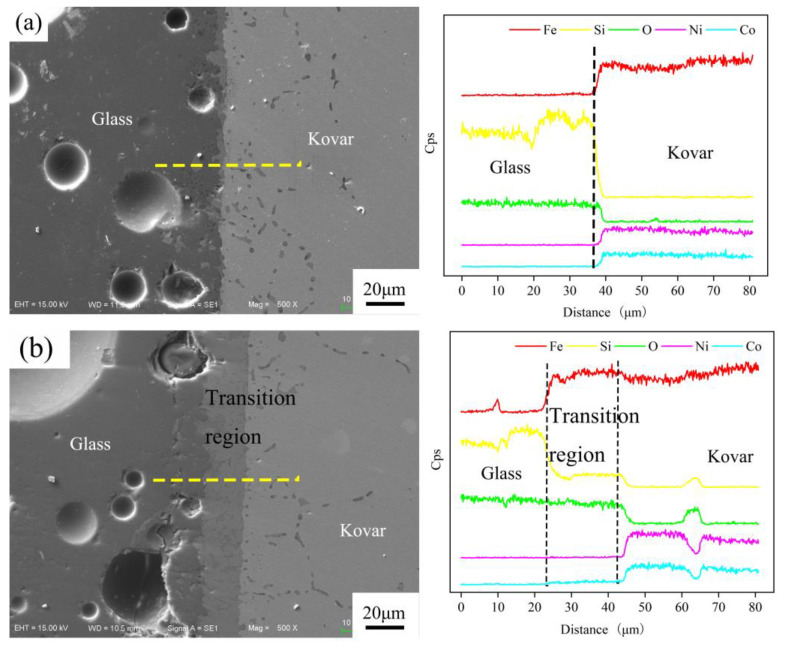
EDS line pattern of borosilicate glass wetting with Kovar: (**a**) no oxidation; (**b**) preoxidation.

**Figure 15 materials-16-04628-f015:**
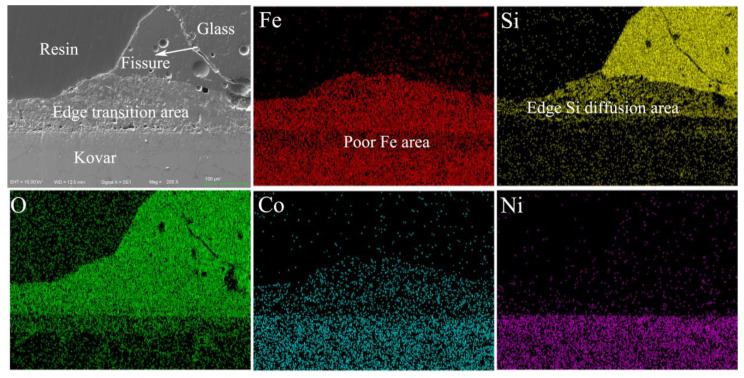
Morphology and map of borosilicate glass and Kovar wetting edge.

**Figure 16 materials-16-04628-f016:**
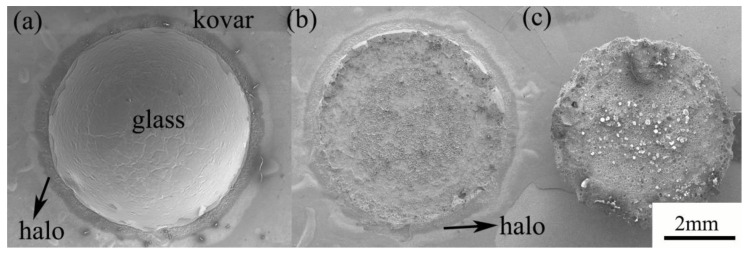
SEM morphology and fracture of borosilicate glass–Kovar wetting. (**a**) Top view of wetted part; (**b**) metal side fracture profile; (**c**) glass side fracture profile.

**Figure 17 materials-16-04628-f017:**
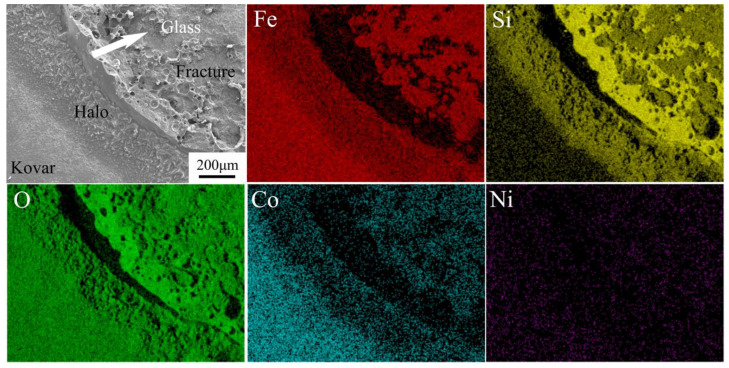
Fracture edge morphology and EDS spectrum.

**Figure 18 materials-16-04628-f018:**
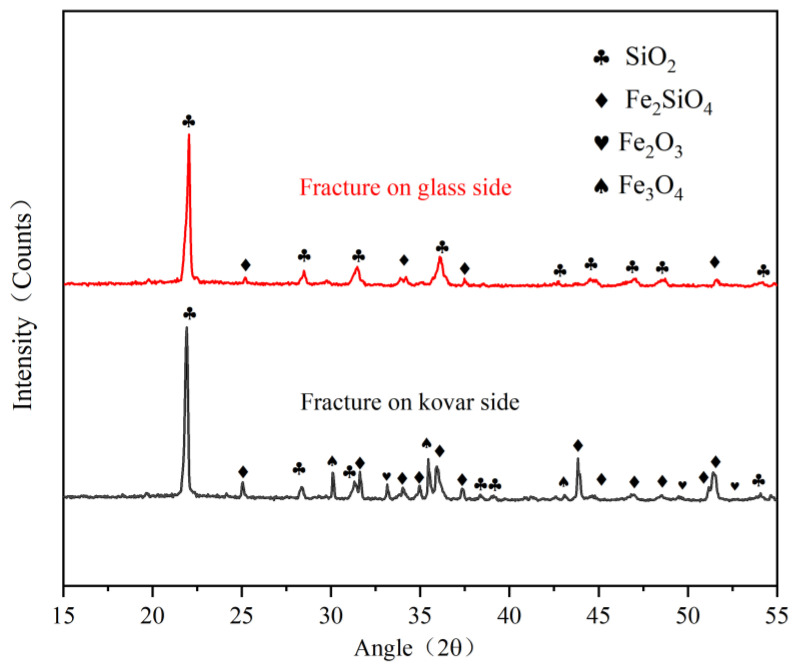
XRD pattern of fracture of Kovar and borosilicate glass.

**Figure 19 materials-16-04628-f019:**
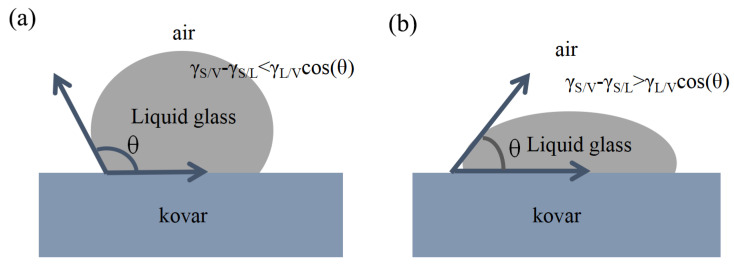
Schematic diagram of wetting: (**a**) not wetting; (**b**) wetting.

**Figure 20 materials-16-04628-f020:**
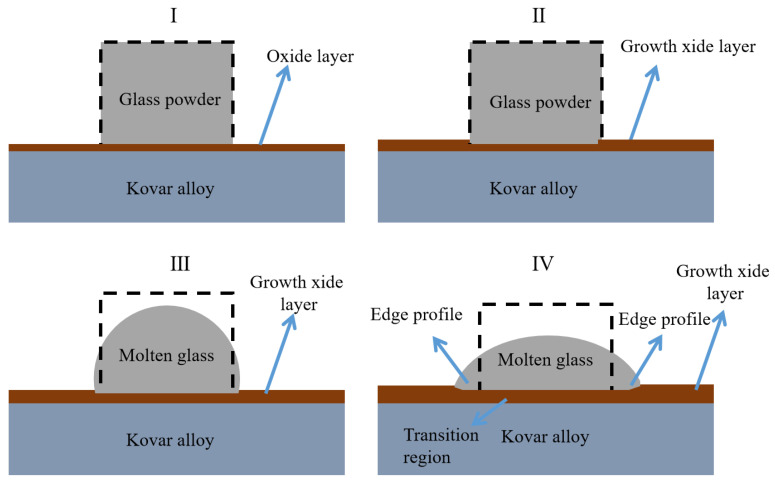
Schematic diagram of the wetting pattern of high-borosilicate glass with Kovar at different stages. (**I**): Wetted initial state. (**II**): Glass unmelting stage. (**III**): Glass melting stage. (**IV**): Wetting final state.

**Figure 21 materials-16-04628-f021:**
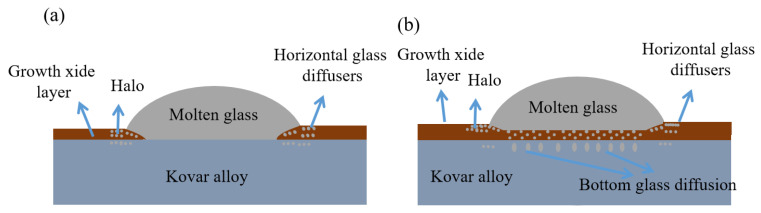
Schematic diagram of the wetting and microscopic diffusion of different treated Kovar alloys with high-borosilicate glass: (**a**) no oxidation; (**b**) preoxidation.

**Table 1 materials-16-04628-t001:** Chemical composition of Kovar alloy (4j29).

Element	Fe	Co	Ni	C, Mn, Si, Cu, Cr, Mo
Composition in wt.%	52.5	17.27	28.78	1.45

**Table 2 materials-16-04628-t002:** Chemical composition of borosilicate glass.

Element	SiO_2_	B_2_O_3_	(Na, K)_2_O	Al_2_O_3_
Composition in wt.%	81	13	4	2

## Data Availability

Not applicable.

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
