# Peer review of "Microscopic Analysis of the Wetting Morphology and Interfacial Bonding Mechanism of Preoxidised Kovar Alloys with Borosilicate Glass"

_materials, 2023, doi:10.3390/ma16134628_

Round 1
Reviewer 1 Report
- The article seems worthy of attention. The article examines the wettability of covarium alloy with high borosilicate glass and microscopically analyzes the mechanism of wettability and diffusion between covarium and borosilicate glass.
- However, a few changes need to be made
- - what it means in the abstract "Firstly, kovar was oxidised at 800℃ for 0 min (..) to observe (...) (0 min?). The same in part 3 experiment. please explain
Introduction: Please expand on how the thickness, composition and microstructure of the oxide influences quality on the metalto-glass sealing bond .
-Table 1 for C,Mn,Si,Cu,Cr,Mo should be value 1,45 (as the sum for these elements)
- (..) and the elemental distribution in the borosilicate glass/kovar bond transition region was analysed using (EDS) " should be probably as "(...) and the elemental distribution in the borosilicate glass/kovar bond transition region was analysed using energy dispersive spectroscopy (EDS)"
- Most of the cited literature is from before 2012, please cite more recent work in this area
Reviewer 2 Report
The work adresses well the industrially important issue of bonding between glass and metal. The methods are well chosen and applied. It is interesting to observe the evolution of the glass powder with temperature and to correlate the microscopy images (drop shape and top views) with EDX material analysis.
The introduction and conclusions are well written, the part of the presentations of results needs major revision to be better understandable.
Observations in view of a revision of the manuscript:
Fig1 (and several others): invert the direction of the arrow between text and image or just pplace a line. As a convention arrows should generally point from the text to the object (at least it's what most authors do, given that then the attention is given to the object/image!).
Fig3: the arrows in the pictures are confusing here (remove arrows to avoid confusion), the contrast of the regions and the words are self-explaining. At present one tries to see if there is something special where the arrows point to, however it's the entire region, not a specific spot, which is discussed).
Figure caption of Fig 3: typo "ofd" => "of" (etc in other places of the manuscript => please make a carefull spell check for the editing!!!)
Fig5: adapt the layout, such that the entire figure is on the same page. It makes the understanding easier, possibly insert also the time directly in the images or at the left per line of images, temperatures for the columns.
From page 7 on, essentially until page 9, several sentences are very difficult to understand (subjects or verbs missing, very long sentences). => please rephrase them carefully (improve the english and make sure that the message is clear).
e.g. "If the temperature increased..." ??? (sentence not clear); "From Fig 8a can be seen..."; "When the temperatire is certain..." ??? (what do you mean?, is there a critical temperature?) etc.
Figure 9: start numbering of the pictures again with a) not with j) [or combine directly with the related figure in a single figure, if appropriate)
"3.3. Discuss" => "3.3 Discussion"
Overall a good work, that is of interest for the fiedl of glass to metal bonding. In summary, the manuscript needs essentially some revision of the text and some of the figures.
see above
Round 2
Reviewer 1 Report
The authors have sufficiently responded to my comments. I believe that the paper should be accepted for publication.